# Revising the Role of Cortical Cytoskeleton during Secretion: Actin and Myosin XI Function in Vesicle Tethering

**DOI:** 10.3390/ijms23010317

**Published:** 2021-12-28

**Authors:** Weiwei Zhang, Christopher J. Staiger

**Affiliations:** 1Department of Biological Sciences, Purdue University, West Lafayette, IN 47907, USA; 2Department of Botany and Plant Pathology, Purdue University, West Lafayette, IN 47907, USA; 3Center for Plant Biology, College of Agriculture, Purdue University, West Lafayette, IN 47907, USA

**Keywords:** plant exocytosis, actin cytoskeleton, microtubules, cellulose synthase, vesicle trafficking, cell wall biosynthesis

## Abstract

In plants, secretion of cell wall components and membrane proteins plays a fundamental role in growth and development as well as survival in diverse environments. Exocytosis, as the last step of the secretory trafficking pathway, is a highly ordered and precisely controlled process involving tethering, docking, and fusion of vesicles at the plasma membrane (PM) for cargo delivery. Although the exocytic process and machinery are well characterized in yeast and animal models, the molecular players and specific molecular events that underpin late stages of exocytosis in plant cells remain largely unknown. Here, by using the delivery of functional, fluorescent-tagged cellulose synthase (CESA) complexes (CSCs) to the PM as a model system for secretion, as well as single-particle tracking in living cells, we describe a quantitative approach for measuring the frequency of vesicle tethering events. Genetic and pharmacological inhibition of cytoskeletal function, reveal that the initial vesicle tethering step of exocytosis is dependent on actin and myosin XI. In contrast, treatments with the microtubule inhibitor, oryzalin, did not significantly affect vesicle tethering or fusion during CSC exocytosis but caused a minor increase in transient or aborted tethering events. With data from this new quantitative approach and improved spatiotemporal resolution of single particle events during secretion, we generate a revised model for the role of the cortical cytoskeleton in CSC trafficking.

## 1. Introduction

In eukaryotic cells, secretion is a fundamental process for delivery of newly synthesized materials to the plasma membrane (PM) or extracellular space to achieve various biological functions. The plant secretory trafficking network is of vital importance in many growth and development processes, including cell morphogenesis, cell wall biosynthesis, hormone signaling, cell polarity formation, as well as response to biotic and abiotic stress [1,2,3,4]. Proteins or lipids are packaged in membrane-bound vesicles that are targeted to specific PM domains for exocytosis. Vesicle trafficking and delivery are highly-choreographed multistep processes requiring a diverse set of proteins and protein–protein interactions to ensure precise spatiotemporal regulation of cargo secretion [5,6]. Typically, secretory vesicles are transported to their destination membrane along cytoskeletal tracks and powered by motor proteins. In some cell types, vesicles are trapped or captured by a cortical cytoskeletal meshwork. Once at the PM, secretory vesicle tethering, docking, and membrane fusion occur in sequential fashion. During tethering, proteins from both the vesicle surface and the PM form a complex to stabilize vesicles near the secretion site; another set of proteins including soluble *N*-ethylmaleimide sensitive factor attachment protein receptors (SNAREs) and SNARE-related proteins further bring vesicles in close proximity during the docking stage; finally, the lipid bilayers of vesicle membrane and PM fuse to allow cargo release and/or incorporation of membrane-associated proteins [7,8,9]. Although the exocytic process and molecular machinery are well characterized in yeast and animal models, emerging studies in plants reveal similar exocytic trafficking systems and a range of evolutionarily-conserved and plant-specific molecular regulators [10,11,12,13].

Plant cells are encased within a carbohydrate-rich cell wall comprised mainly of cellulose microfibrils as the major load-bearing component. Cellulose is a β-1,4-linked glucan chain that is synthesized by cellulose synthase (CESA) enzymes located at the PM and multiple CESAs are arrayed in large, multimeric rosette-shaped cellulose synthase complexes (CSCs) [14]. CSCs are thought to be assembled in the Golgi and delivered to the PM by post-Golgi vesicles [15,16]. Some of the CSC-containing vesicles are referred to as small CESA compartments (SmaCCs) or microtubule-associated CESA compartments (MASCs) [17,18]. CSCs are preferentially delivered to PM sites that coincide with cortical microtubules [18]. Furthermore, CSCs at the PM are linked to underlying cortical microtubules via microtubule-associated proteins such as CELLULOSE SYNTHASE INTERACTIVE 1/POM2 (CSI1/POM2) and move in linear trajectories at a constant speed to produce cellulose microfibrils [19,20,21,22]. Although cortical microtubules define their trajectories, the motility of CSCs is thought to be driven by the force of glucan chain polymerization [15,19].

Recent advances in high spatiotemporal resolution live-cell imaging and utilization of functional, fluorescent protein-tagged CESAs have allowed CSC trafficking to emerge as a powerful model for studying exocytosis in plant cells [15,18,23,24,25]. In particular, single-particle tracking has illuminated specific steps in the exocytosis process based on the size, unique spatial localization, and motility patterns of CSCs on different endomembrane compartments [15,18,23,24]. For example, putative CSC delivery vesicles arrive in the cortical cytoplasm near the PM, display erratic local movement for several seconds, become spatially restricted in their movement and maintain a static position for 60–90 s, and then undergo steady and linear movement as catalytically active CESA complexes (see [18,23,25] and Figure 1A). The pause phase is hypothesized to encompass tethering, docking, and fusion of CSC vesicles, possibly followed by the activation of CESAs before they synthesize cellulose and show linear motility.

Through genetic studies as well as fluorescent tagging and colocalization analyses, several proteins that transiently associate with CSC particles during the pause phase and may play a role in the efficient delivery of CSCs to the PM have been identified (Figure 1B). Some are exocytosis-related machinery proteins, including the exocyst tethering complex which coappears with CSCs at the beginning of the pause phase for ~12 s, suggesting that vesicle tethering immediately follows the arrival of CSC compartments at the PM [25,26]. The plant-specific protein PATROL1 (PTL1), a homolog of Munc13 proteins in animals with a role in SNARE complex assembly, colocalizes with CSCs for 1–2 s at the pause phase and often appears 1–2 s after the appearance of exocyst subunits [26]. KEULE, a plant homolog of Sec1/Munc18 proteins implicated in regulating SNARE function, also arrives at the beginning of the pause phase, has an average lifetime of 3.7 s, and is likely to play a role in docking and/or fusion of CSC vesicles [27]. The actin-based motor, Myosin XIK, coappears with cortical CSC compartments and remains present at the beginning of the pause phase for 3–9 s before disappearing [23,25]. Finally, a new plant-specific protein, TRANVIA (TVA), arrives at the PM with CESA delivery compartments and remains associated with CSCs during the pause phase for an average of 6.3 s, and likely plays a role in CSC delivery [24]. The tightly regulated appearance and timing of these molecular players during the initial vesicle tethering phase suggest that the pause phase is critical for complex molecular interactions that ensure successful and efficient CSC delivery to the PM. Despite the identification of several candidate proteins, the precise molecular mechanisms and sequence of events that occur during the pause phase and the details of regulation of tethering, docking, and fusion remain to be fully elucidated.

The actin cytoskeleton and myosin motors are key contributors to long distance intracellular transport in plant cells [28,29,30,31,32]. The global distribution of CSC-containing Golgi in the cortical cytoplasm relies on an intact actin cytoskeleton network and genetic or pharmacological disruption of actin reduced the exocytosis rate of CSCs at the PM [18,33]. Within the myosin XI family, the XIK isoform is the major isoform responsible for organelle transport and motility in somatic cells [31,34]. Using CSC trafficking as a model system, we provided evidence that myosin XI participates in vesicle exocytosis near the PM and this process is likely mediated by interactions with the exocyst complex [23,25]. Disruption of myosin XI activity, either genetically or with inhibitor treatment, results in reduced rate of overall delivery of CSCs into the PM, increased failure of late exocytosis events, and altered CSC lifetime during the pause phase [23,25]. Myosin XIK directly interacts with several exocyst subunits and is functionally associated with the exocyst complex at CSC delivery sites [25]. Inhibition of myosin XIK activity results in reduced localization and a shorter lifetime of exocyst complex subunits at the PM during CSC delivery [25]. These results indicate a role for myosin XI in the initial vesicle tethering stage; however, the exact mechanism and which sub-steps require the function of myosin remain unclear.

Despite evidence that both the actin and microtubule cytoskeletons are implicated in CSC secretion at the PM [16,35,36], the exact roles they play during exocytosis remain to be fully elucidated. Here, we develop a quantitative imaging approach to directly measure the pause or tethering frequency of CSC compartments at the PM and our results confirm that actin and myosin XI are involved in the initial tethering of vesicles during CSC exocytosis. In addition, treatments with the microtubule inhibitor, oryzalin, discount a primary role for cortical microtubules in vesicle tethering and allow us to revise and expand current models for the role of cortical cytoskeleton in the last steps of CSC secretion.

## 2. Results

### 2.1. Myosin XI and Actin Play a Key Role in Successful Insertion of CSCs at the PM

As previously described, using high resolution spinning-disk confocal microscopy (SDCM) and live cell imaging, individual CSC exocytosis and insertion events can be visualized and quantified through analysis of kymographs of particles at/or near the PM [18,23,25]. Previously, we found that the majority (~90%) of CSC insertion events in wild-type epidermal cells were successful and characterized by a pause phase (magenta dashed line) followed by a steady movement phase (green dashed line); however, a small proportion (~10%) of events only displayed a pause phase without the steady movement phase, suggesting a failed/aborted exocytosis event [23,25] (Figure 1C). Quantifying the frequency of failed insertion events revealed that there was a two- to five-fold increase in failed events in myosin XI- or actin-deficient cells compared with that in wild type, supporting a role for actin and myosin in mediating exocytosis of CSCs. Moreover, the increased CSC insertion failures correlated with increased frequency of events that had abnormal duration of the pause phase (either shorter or longer) in myosin XI- or actin-deficient cells [23,25].

To validate previously published results, we performed the single-particle CSC insertion analysis on myosin *xik-2* mutant, wild-type cells treated with the myosin inhibitor pentabromopseudilin (PBP) [23], or on wild-type cells treated with the actin polymerization inhibitor latrunculin B (LatB). The Arabidopsis *myosin xik-2* and wild-type sibling lines expressing YFP-CESA6 in a *prc1-1/cesa6* homozygous background were described previously [25] and used for quantitative live-cell imaging with spinning-disk confocal microscopy (SDCM). Dark-grown seedlings were pretreated with either mock or inhibitor solution for 10 min prior to imaging of hypocotyl epidermal cells with acquisition at 3 s intervals for 10 min. Individual CSC insertion events analyzed by kymograph were grouped into five categories based on whether the insertion was successful and/or whether they had a different duration of the pause phase (Figure 1C,D). The mean pause time in mock-treated wild-type cells was 81 ± 31 s (mean ± SD, n = 87 events), consistent with our previously reported values [23,25]. A shorter or longer pause was defined as any event outside the mean pause time of wild type plus or minus 1 SD (<50 s or >112 s). The results showed that the frequency of failed CSC insertion events was only 12% in mock-treated wild-type cells, whereas in mock-treated *xik-2* or wild-type cells treated with LatB the frequency increased by ~2.5-fold, and in wild-type cells treated with PBP it increased by 4-fold (Figure 1D).

Analysis of the duration of the pause phase for each single-particle insertion event revealed that the population of short pause events (<50 s) increased from 11.5% in wild type to 18%, and 30% and 20% in *xik-2*, PBP- and LatB-treated cells, respectively (Figure 1E). In addition, the events with long pauses (>112 s) increased from 17% in wild type to 27%, and 30% and 21% in *xik-2*, PBP- and LatB-treated cells, respectively (Figure 1E). It should be noted that the greater inhibitory effect observed in PBP- and LatB-treated cells compared with our previous report [23] is likely due to the use of a 10 min pretreatment of seedlings with inhibitors, whereas in previously published assays seedlings were mounted directly in inhibitor solutions without pretreatment. In addition, PBP treatment resulted in greater inhibition of CSC insertion at the PM compared with that in *xik-2* cells, suggesting that PBP targets multiple myosins that are functionally redundant with XIK during CSC secretion.

These results confirm that actin and myosin XI play a major role in the successful and efficient insertion of CSCs at the PM. Combined with our previous finding that myosin XIK directly interacts with the exocyst complex and transiently colocalizes with exocyst subunits for the first 3–9 s of the pause phase [25], we propose that myosin XI and actin play a role in the vesicle tethering step during CSC exocytosis.

### 2.2. The Frequency of Vesicle Tethering Is Reduced in Myosin- or Actin-Deficient Cells

Although the evidence above implicates actin and myosin XI in regulating vesicle tethering during CSC insertion at the PM, a direct quantification of vesicle tethering frequency to support this argument is necessary. The single-particle CSC insertion approach described above does not provide a direct measure of vesicle tethering, nor does it elucidate the exact trafficking step that actin or myosin XI affect, because the altered pause phase or failed insertion observed in cells could be due to a defect in vesicle tethering, to normal tethering, but inefficient docking and/or fusion, or both. Mutation of *Arabidopsis* KEULE, a plant homolog of the Sec1/Munc18 protein that is involved in regulating SNARE function and vesicle docking/fusion [37], causes an overall reduction in CSC delivery rate and a longer pause phase during CSC insertion; however, the CSC tethering frequency appears to be unaffected, indicating a role for KEULE only in vesicle docking and/or fusion [27].

Here, we revised the single-particle tracking approach described by Gutierrez [27] to quantitatively measure CSC tethering frequency, in order to provide additional evidence that actin and myosin XI are required for vesicle tethering and to gain better understanding of the spatiotemporal regulation of vesicle secretion in plant cells. The SDCM timelapse series collected at 3 s intervals for 10 min and shown in Figure 1 were used for tethering frequency analysis. To estimate tethering events per unit area and time, a region of interest (ROI) with a size of 60 × 60 pixels^2^ box (63.68 µm^2^) was selected and all tethering events within the ROI during a 6 min time period were measured by analysis of kymographs (Figure 2A,B). Based on previous evidence from our group and others [25,26], a tethering event is predicted to occur at the beginning of the pause phase, which is indicated as a continuous vertical line on kymographs. Previous analyses showed that the duration of the pause phase ranges from 30 s to 2 min or longer. In this assay, any vertical and continuous line that lasted for at least 30 s was counted as a vesicle tethering event regardless of whether subsequent insertion occurred. To aid in the detection of vertical tracks on kymographs, time series were adjusted with two-frame averaging using a Grouped Z Project function.

The average tethering frequency in wild-type cells treated with mock solution was 0.22 ± 0.01 events µm^−2^ min^−1^ (Figure 2C). Following genetic or chemical inhibition of myosin XI activity, there was a ~20% reduction in tethering frequency in *xik-2* cells and a ~50% reduction in wild-type cells treated with PBP for 10 min (Figure 2C). Inhibiting actin function with short-term LatB treatment had an intermediate effect with a ~37% reduction in tethering frequency (Figure 2C). It should be noted that the CSC tethering frequency in wild-type cells measured by this assay was higher than the estimated CSC delivery rate measured using FRAP (0.11 ± 0.02 events µm^−2^ min^−1^) [23]; this could be because there is a proportion of tethered vesicles that fail to deliver CSCs at the PM, other types of pause events of CSCs occur at the PM which are not related to delivery or secretion, or both.

Because only CSCs that remain stationary at the PM for at least 30 s were included in the tethering frequency analysis, there is a potential that some transient and aborted tethering events were missed; although, it is unknown whether such transient behavior of CSCs occurs frequently in cells. To address this question, we quantified the frequency of CSC pause events with a duration of less than 30 s in wild-type, *xik-2* or inhibitor-treated cells. Time-lapse images collected at higher temporal resolution of 1-s intervals were used to aid in the detection of short tethering/pause events on kymographs. Our results revealed that the frequency of the pause events with duration <30 s in wild-type cells was 0.022 ± 0.002 events µm^−2^ min^−1^ (Figure 2D), which is ~10 times lower than the frequency of tethering events that lasted for 30 s or longer in wild-type cells (Figure 2C). In addition, no significant differences were observed between wild-type, *xik-2*, or PBP- and LatB-treated wild-type cells (Figure 2D). These results indicate that a vesicle pause event, or tethering at the PM, generally lasts for >30 s and that aborted tethering events associated with CSC insertion are relatively uncommon. The significantly reduced CSC tethering frequency observed in myosin *xik* or myosin and actin inhibitor-treated cells confirmed that myosin XI and actin are required for efficient vesicle tethering during CSC secretion.

### 2.3. Cortical Microtubules Play a Minor Role in CSC Vesicle Tethering and Fusion

Cortical microtubules are undoubtedly key players in CSC delivery or recycling by mediating the interaction with SmaCCs/MASCs and predicting the PM insertion sites for newly arrived CSCs [17,18,26]. Under normal conditions, 78% of successful CSC insertion events are observed to occur at PM sites that are coincident with cortical microtubules [18]. Using transgenic plants co-expressing YFP-CESA6 and the microtubule reporter mCherry-TUA5 [18], we also observed that 85% of the CSC particles (53 out of 62 events) were delivered to cortical microtubule sites and this association occurred coincident with the start of the pause phase (Figure 3A; Appendix A). Moreover, the frequency of co-occurrence with microtubules was unaffected in cells treated with actin or myosin inhibitors [23,33], suggesting that the targeting of CSCs to microtubules is independent of actomyosin activity. Although microtubules correlate with CSC insertion sites, possibly via the CESA-microtubule linker protein CSI1 [26], disruption of cortical microtubules with the inhibitor oryzalin or genetic mutation of CSI1 does not affect the overall delivery rate or abundance of CSCs at the PM [18,19,21]. These findings question an absolute requirement of cortical microtubules in CSC delivery or insertion at the PM. In addition, it is unclear whether cortical microtubules participate in specific exocytosis steps including vesicle tethering, docking or fusion. To determine whether microtubules are involved in specific steps of CSC delivery and secretion at the PM, we performed single CSC insertion and tethering frequency analysis on cells treated with the microtubule inhibitor oryzalin.

To substantially deplete microtubules from the cell cortex, Arabidopsis seedlings expressing YFP-CESA6 were pretreated with 20 µM oryzalin for 2 h prior to SDCM imaging. The effect of inhibitor treatment was verified by examining the microtubule reporter line YFP-TUB5 [38] under the same treatment conditions (Figure 3B). In oryzalin-treated cells, CSC particles were still present at the PM with a similar overall abundance but altered distribution pattern (Figure 3B,D), as reported previously [19]. Cytoplasmic vesicles are associated with CSC delivery to the PM and we previously demonstrated that inhibition of actin or myosin XI activity caused accumulation of CESA-containing vesicles in the cortical cytoplasm [23]. Because cortical microtubules also interact with cytoplasmic CESA compartments and position their PM delivery sites [18], we next tested whether depletion of cortical microtubules would alter the abundance of cytoplasmic CESA vesicles. As described previously [23], we analyzed vesicle density in both the cortical (0–0.4 µm below the PM) and subcortical (0.6–1 µm below the PM) cytoplasm and no difference was observed in either region in oryzalin-treated cells compared to mock-treated cells (Figure 3C,E). This suggests that localization of CSC vesicles to the cell cortex does not require cortical microtubules.

Because the targeting of CSCs to cortical microtubules appears to be coincident with tethering of CSCs to the PM (both occur at the beginning of the pause phase), it is possible that microtubules are required for efficient tethering/docking of CSCs. To test this possibility, we performed single-particle CSC insertion assays as well as CSC tethering frequency analysis on oryzalin-treated cells. By tracking individual CSC insertion events and analysis of kymographs, we observed that in oryzalin-treated cells, 89% of the tethering events observed were successful in delivering a functional CSC, and this was not significantly different from the value of 87% in mock-treated cells (Figure 4A,D). In addition, analysis of CSC tethering frequency showed that oryzalin-treated cells were not significantly different from mock-treated cells (Figure 4B). These results suggest that the normal tethering and fusion of CSC vesicles at the PM is not affected in cells with cortical microtubule arrays disrupted. However, aborted CSC tethering events (<30 s pause) increased by 20% in oryzalin-treated cells compared with mock-treated cells (Figure 4C), suggesting that cortical microtubules play a minor role in establishing stable tethering of CSC vesicles to the PM during secretion. Moreover, a similar increase in the frequency of aborted tethering events was observed in cells treated with oryzalin for 2 h and then PBP for 10 min, but not in cells treated with PBP alone (Figure 4C), suggesting that the role of microtubules is either independent or upstream of the actomyosin-mediated tethering process. Finally, we observed that in cells treated with oryzalin the standard pause event time (based on mean ± 1SD in wild type; 51–115 s) was modestly but significantly increased to 85 ± 12 s compared to 78 ± 17 s in wild type (Figure 4E).

These data indicate that cortical microtubules may influence the CSC pause time during PM insertion; although, the detailed mechanism is unknown. Collectively, these results indicate that cortical microtubules do not play a significant role in CSC vesicle delivery or the efficiency of tethering and fusion, but may simply serve as a landmark for vesicle capture in the cortical cytoplasm (Figure 5).

## 3. Discussion

Recent advances using single-particle tracking of CSC delivery events at the PM along with genetic and small molecule inhibitor approaches reveal several evolutionarily-conserved and plant-unique players as well as specific molecular steps during late stages of the secretory processes. In this study, we developed a quantitative image analysis approach to measure vesicle tethering frequency at the PM using single-particle CSC delivery as a surrogate for exocytic events and provide direct evidence that actin and myosin XI are required for the initial vesicle tethering step during exocytosis. Further, by quantitatively assessing the late stages of CSC secretion in oryzalin-treated cells, it becomes obvious that cortical microtubules only play a minor role in vesicle tethering, docking, or insertion even though they serve as a landmark for the beginning of the pause phase (Figure 5).

The evidence that myosin and actin play a key role during the final stages of CSC delivery is expanding rapidly. The reduced vesicle tethering rate observed here for *myosin xik* as well as actin- and myosin-inhibitor treated cells is consistent with previous findings of a reduced rate of CSC delivery in *myosin xi*, *actin2/7*, and upon acute actin- and myosin-inhibitor treatment [23,33]. Moreover, abnormal accumulation of CESA compartments in the cortical cytoplasm adjacent to the PM was observed when actin or myosin function are perturbed, and is a typical phenotype of cells defective in vesicle tethering and/or fusion [23,25]. The accumulation of cortical CESA vesicles in actin- or myosin-deficient cells further indicates that the reduced vesicle tethering frequency is unlikely caused by a reduction in cytoplasmic streaming, which would prevent the access or delivery of vesicles to the cell cortex prior to tethering at the PM. Moreover, the timeline for arrival and disappearance of proteins at the exocytosis site can be indicative of their function during exocytosis. Previous work demonstrated that myosin XIK appearance at the CSC tethering site overlaps with the exocyst complex during the first 3–9 s of the pause phase, further supporting a role for myosin in the initial vesicle tethering step [25]. Critically, both the lifetime and abundance of stationary exocyst complex subunits depend upon myosin XI and actin function [25]. How myosin XI and actin coordinate with the exocyst complex and other potential players to mediate vesicle tethering requires additional investigation; however, actomyosin function could be required for the efficient recruitment and stabilization of exocyst complex to the vesicle tethering site, as suggested previously [25].

Our latest results confirm a role for actin and myosin XI in the vesicle tethering step of exocytosis; nevertheless, we cannot exclude the possibility that actin or myosin XI are also involved in subsequent steps of exocytosis such as docking or membrane fusion. Myosin V, the closest homolog of myosin XI in animal systems, interacts with SNARE proteins and assists vesicle docking at the PM [39]. The timing of association of myosin XIK with CSCs during the pause phase appears to partially overlap with that of PTL1 and KEULE, two proteins that are implicated in SNARE function and play a role in CSC vesicle docking and/or fusion [26,27]. PTL1 and KEULE show transient colocalization with CSCs during the early pause phase for only 2–4 s. The newly discovered TVA protein also colocalizes with CSC secretory vesicles and has a similar behavior to myosin XIK; it arrives with the cortical vesicle during the erratic phase and remains associated with the pause phase for ~6 s before departing [24]. These findings suggest that TVA is another regulator of vesicle tethering and may indeed cooperate with myosins or exocyst during this key step in secretion. Interestingly, fluorescently labeled TVA moves away from the CSC tethering site as discrete particles after insertion of CSCs at the PM, resembling a kiss-and-run vesicle fusion event [24], indicating that the molecular processes choreographing CSC secretion during the early pause phase are more complex and may require distinct mechanisms when compared with conventional exocytosis. Actin and myosin have been shown to regulate fusion pore dynamics during membrane fusion including kiss-and-run fusion events in animal exocrine cells [40]. Further studies to dissect the role of the actin cytoskeleton and myosin XI in the docking and fusion processes as well as identification of new molecular players will help uncover the regulation of exocytosis in plant cells as well as advance our understanding of plant growth and development in general.

Cortical microtubules are key players in plant cell wall biosynthesis and are prominently featured in most models describing the final stages of CSC delivery to the PM [15,35,41]. These models are supported by observations that 80–90% of CSCs pause on cortical microtubules prior to successful PM insertion events [18,21,23]. However, the targeting of CSC compartments to cortical microtubules during PM insertion seems not to be a rate-limiting step, because other data show that depletion of microtubules with oryzalin or genetic mutation of the linker protein CSI1 does not affect the overall CSC delivery rate measured by FRAP [18,21,26]. In this study, we performed CSC tethering analysis as well as single-particle CSC insertion assays to investigate the role of cortical microtubules at high spatiotemporal resolution and test directly whether cortical microtubules play a role, perhaps in coordination with actin and myosin, in facilitating efficient vesicle tethering and docking during CSC insertion. Our results showed that depletion of cortical microtubules with oryzalin did not significantly affect either the abundance of cortical CSC vesicles or the subsequent PM tethering, docking, or fusion steps during CSC exocytosis. Nevertheless, cortical microtubules may play a minor role in maintaining a stable PM tethering complex during CSC exocytosis, because we observed a 20% increase in aborted tethering events in oryzalin-treated cells. Therefore, we conclude that the targeting of CSCs to cortical microtubules is not an essential step for CSC insertion at the PM nor is it necessary for actomyosin-mediated tethering, docking and fusion steps during secretion. Perhaps the capture of CSCs on cortical microtubules simply ensures that successfully inserted CSCs are ready and able to translocate along cortical microtubules and assures that the trajectory of cellulose microfibrils in the wall is dictated by the orientation of cortical microtubule arrays (Figure 5). Mechanistically, this could either be a redundant step or a failsafe to ensure that orientated plant cell wall deposition is primed to follow an actomyosin-dependent delivery event. We propose a revised model that PM tethering and exocytosis processes of CSC vesicles are mainly mediated through cortical actomyosin-dependent function, whereas cortical microtubules serve as a landmark to position CSC insertion sites at specific locations on the PM (Figure 5). Although the capture on microtubules and the PM tethering of CSCs occurs roughly at the same time, the exact order of the two molecular events remains to be resolved. Quantitative, high-resolution colocalization assays with markers of microtubules, exocyst, or myosin XI as well as genetic studies with mutants of these components are required to address this question.

The involvement of both microtubule and actin cytoskeletons at PM-associated secretion sites supports prior evidence that cortical microtubules and actin filaments are often colocalized in rapid elongating cells and their organization and dynamics are interdependent [42]. In particular, actin filament “pausing” events were frequently observed in the proximity of cortical microtubules in untreated cells and, during recovery from LatB treatments, short actin filament fragments coaligned and translocated along microtubules. It is feasible that such colocalization sites represent microtubule-defined exocytosis sites; although, it remains unclear what upstream signaling events determine their location or what molecular cues recruit secretory machinery proteins as well as microtubule- and actin-related cytoskeletal elements to these secretion sites. The function of microtubules as CSC secretion landmarks is likely to be associated with the linker protein CSI1; however, the role of CSI1 in CSC secretion needs to be further investigated. One study suggests that CSI1 localized to insertion sites many seconds prior to the arrival of CSCs [26]; however, another report shows that only 23% of the newly-delivered CSC particles co-associate with CSI1 fluorescent puncta and the frequency of CSC insertion on cortical microtubule sites is unaltered in *csi1/pom2* mutants [21].

At a broader level, our findings indicate an evolutionarily-conserved mechanism is shared among diverse eukaryotes and demonstrate that the late stages of exocytosis are mainly dependent on the actin cytoskeleton and myosin motors rather than microtubules.

## 4. Materials and Methods

### 4.1. Plant Materials and Growth Conditions

An *Arabidopsis thaliana* Col-0 line expressing YFP-CESA6 in the homozygous *prc1-1/cesa6* background [19] was kindly provided by Ying Gu (Pennsylvania State University). Homozygous *myosin xik-2* and wild-type siblings expressing YFP-CESA6 *prc1-1* were recovered from an F3 population resulting from the cross between a *myosin xi1*, *xi2*, *xik* triple knockout (*xi3KO*) mutant and the YFP-CESA6 *prc1-1* line as described previously [23,25]. Transgenic plants expressing YFP-TUB5 were described previously [38]. The YFP-CESA6 mCherry-TUA5 co-expression line was kindly provided by David W. Ehrhardt (Carnegie Institute for Science).

Arabidopsis seed was surface sterilized and stratified at 4 °C for 3 d on half-strength Murashige and Skoog (MS) medium supplemented with 0.8% agar. For light growth, plants were grown under a light intensity of 120–140 µmol m^−2^ s^−1^ under long-day conditions (16 h light/8 h dark) at 21 °C. For dark growth, plates were exposed to light for 4 h and then placed vertically and kept at 21 °C for 3 d in continuous darkness.

### 4.2. Live-Cell Imaging

Epidermal cells from the apical region of 3-day-old dark-grown hypocotyls were imaged by spinning-disk confocal microscopy (SDCM). Image acquisition was performed using a Yokogawa scanning unit (CSU-X1-A1; Hamamatsu Photonics, Hamamatsu, Japan) mounted on an Olympus IX-83 microscope, equipped with a 100× 1.45 numerical aperture (NA) UPlanSApo oil objective (Olympus America Inc., Waltham, MA, USA) and an Andor iXon Ultra 897BV EMCCD camera (Andor Technology, Concord, MA, USA). YFP fluorescence was excited with a 514 nm laser line and emission collected through a 542/27 nm filter. For cortical and subcortical YFP-CESA6 imaging, z-series at 0.2 µm step sizes plus time lapse with 2 s intervals for 10 frames were collected.

### 4.3. Image Processing and Quantitative Analysis

Image processing and analysis were performed with Fiji Is Just ImageJ [43].

A high spatiotemporal resolution single-particle CSC insertion assay was performed as described previously [23]. Time series collected by SDCM at 3 s intervals for 10 min were used for this analysis. Only CSC particles that showed de novo appearance at the plane of the PM followed by a pause phase of more than 5 frames (>15 s) were considered to be new insertion events. The presence and duration of the pause phase was determined by analysis of kymographs. A line was drawn along the trajectory of a newly inserted CSC particle and a kymograph was generated with the Multi Kymograph function in FIJI. A straight vertical line on the kymograph was considered to be a pause event. The duration of the particle pause phase was determined by fitting a straight line along the path of continuous movement and another line along the pause phase on the kymograph. The intersection of the two lines was defined as the end of the pause phase. For quantification, 10 random insertion events were analyzed in each cell and a total of 10–12 cells were measured from at least 5 seedlings per genotype or treatment.

To analyze CSC tethering frequency, the same SDCM time series used for the single-particle insertion assay collected at 3 s intervals for 10 min were used. To estimate a tethering frequency (number of events per unit area and time), a region of interest (ROI) with a size of 60 × 60 pixels^2^ box (63.68 µm^2^) at the PM was chosen and all tethering events were measured within the ROI during a 6 min time span. Kymograph analysis was used to quantify tethering events. To aid in detection of tethering events on kymographs, 2-frame averaging was applied to the time series with the Group Z Project function in FIJI using the average intensity method. The time series were pre-rotated to make sure that most of the CESA trajectories were fitting vertical lines which is the same orientation that all the kymographs were generated. To create kymographs that cover all of the particle trajectories in the ROI, the Reslice function in FIJI was used and “start at left” was chosen. All vertical lines on kymographs that lasted for 30 s or longer were counted as vesicle tethering events regardless of whether they were successful insertions or not. The frequency of tethering was calculated as the number of tethering events per unit area divided by elapsed time. For quantification of transient/aborted tethering events, SDCM timelapse images collected at 1 s intervals for 6 min were used. The same kymograph approach described above was used to identify transient tethering events and only vertical lines that had a duration of longer than 3 s but less than 30 s were counted.

Analysis of PM-localized CSC density and cortical and subcortical CESA compartment density were performed as described previously [23].

### 4.4. Drug Treatment

For short-term live cell treatments, seedlings were submerged in mock or inhibitor solution in a 24-well plate or directly on the slide and kept in the dark prior to mounting and imaging. The myosin inhibitor pentabromopseudilin (PBP; Adipogen, San Diego, CA, USA), actin polymerization inhibitor latrunculin B (LatB; Calbiochem, San Diego, CA, USA), and microtubule depolymerizing agent oryzalin (Sigma-Aldrich, St. Louis, MO, USA) were dissolved in DMSO to generate stock solutions that were stored at −20 °C and diluted in water immediately prior to use.

### 4.5. Statistical Analysis

One-way ANOVA with Tukey’s post-hoc tests were performed in SPSS (version 27) to determine significance among different treatments. Two-tailed Student’s *t*-tests were performed in Excel 16.51. Chi-square tests were used for statistical comparison of data that did not follow parametric distributions and *p* values were calculated in Excel 16.51.

## Figures and Tables

**Figure 1 ijms-23-00317-f001:**
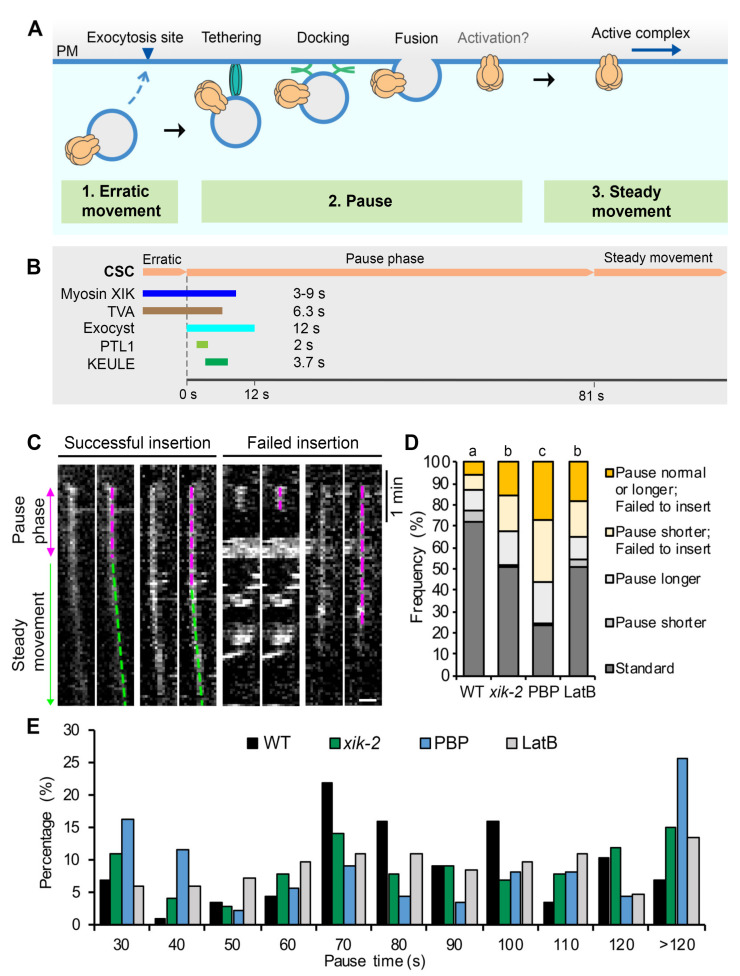
Myosin XI and actin help choreograph the successful insertion of CSCs at the PM: (**A**) A schematic diagram shows a typical CSC insertion event at the PM. A CESA compartment initially appears near the PM and undergoes erratic movement, likely representing a delivery vesicle that is approaching an exocytosis site. The compartment then pauses and exhibits a static or pause phase, likely corresponding to tethering, docking, and fusion of the delivery compartment to the PM. There may also be an activation step for CESA catalytic activity during the pause phase. After a CSC rosette is inserted, it shows steady and linear movement in the PM as an active complex. (**B**) A summary of the proteins reported to be transiently associated with CSCs during PM insertion and a timeline of their duration at the erratic or early pause phase. The time shown for each protein represents the mean lifetime or duration of association from the beginning of the pause phase based on published results (see main text for details and original citations). (**C**) Representative kymographs show successful or failed CSC insertion events with longer or shorter pause times at the PM. Epidermal cells from 3-day-old dark-grown hypocotyls were imaged using spinning-disk confocal microscopy (SDCM) with time-lapse series collected at 3 s intervals. For these experiments, *Arabidopsis thaliana prc1-1/cesa6* homozygous lines expressing functional YFP-CESA6 were imaged. A successful insertion event includes a pause phase (magenta dashed lines) followed by a steady movement phase (green dashed lines), whereas a failed insertion event only has a pause phase. Bar = 1 μm. (**D**) Percentages of the different types of insertion event in the *myosin xik-2* mutant or in inhibitor-treated cells. Three-day-old etiolated hypocotyls of *xik-2* or wild-type (WT) were treated with mock (0.2% DMSO), 10 μM pentabromopseudilin (PBP), or 10 μM latrunculin B (LatB) for 10 min prior to SDCM imaging. The standard insertion process was defined as a successful insertion event that has a pause time within the mean value minus or plus one standard deviation (81 ± 31s) in wild type (WT); any pause <50 s) or >112 s was considered as shorter or longer pause. A total of 87, 99, 86, and 81 insertion events from 10 cells were measured in WT, *xik-2*, PBP, and LatB cells, respectively; chi-square test, *p* < 0.05. (**E**) Distribution of CSC pause times at the PM for the insertion events shown in (**D**).

**Figure 2 ijms-23-00317-f002:**
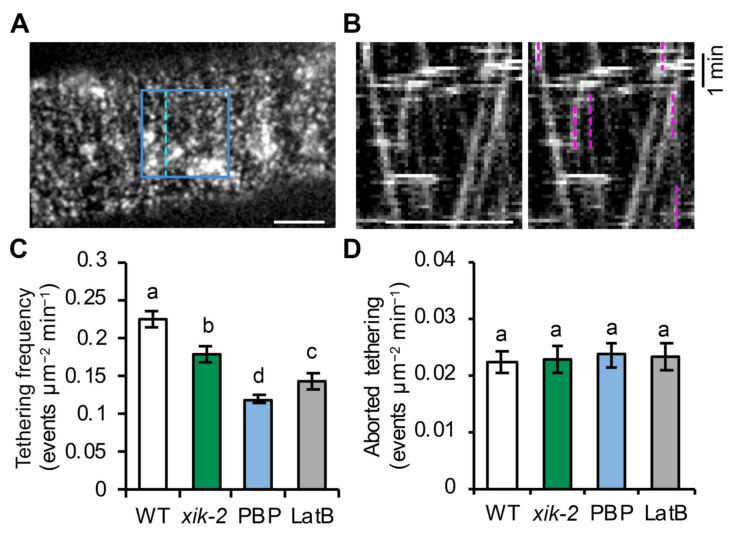
Myosin XI and actin regulate the tethering frequency of CSCs: (**A**) A representative single image shows the PM plane of Arabidopsis hypocotyl epidermal cells expressing YFP-CESA6 imaged with SDCM. A region of interest (blue box) was selected for analysis of tethering frequency. Dashed line indicates where the kymograph shown in (**B**) was generated. Bar = 5 μm. (**B**) A representative kymograph shows the trajectories of CSCs at the PM. Vertical lines (marked with magenta dashed lines) represent the pause phase and were quantified as tethering events, if they were at least 30 s in duration. Bar = 5 μm. (**C**) Quantification of tethering events shows a significant reduction in vesicle tethering frequency in *xik-2*, PBP- or LatB-treated cells compared with that in WT cells. Genotypes and treatments were as described in Figure 1. Values given are means ± SE (n = 10–12 cells per genotype or treatment, one-way ANOVA with Tukey’s post hoc test, letters (a–d) denote samples/groups that show statistically significant differences from other groups, *p* < 0.01). (**D**) Quantification of aborted tethering events with a duration of <30 s shows no difference among genotypes or treatments. Values given are means ± SE (n = 10–12 cells per genotype or treatment, one-way ANOVA with Tukey’s post hoc test, *p* > 0.05).

**Figure 3 ijms-23-00317-f003:**
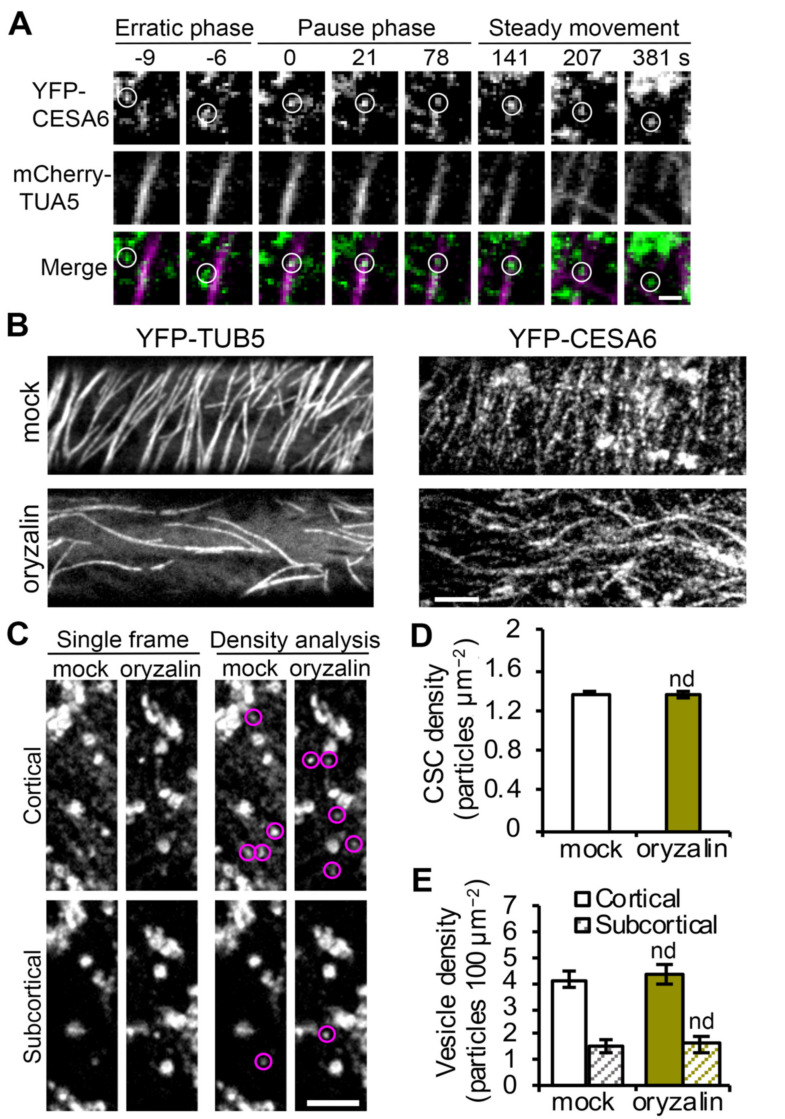
Disruption of cortical microtubules does not alter the abundance of PM-localized CSCs or cortical CESA compartments: (**A**) Representative images from a time series showing a single CSC particle (marked by white circles) insertion event at the PM. SDCM imaging of a transgenic Arabidopsis line co-expressing YFP-CESA6 and mCherry-TUA5 and single CSC insertion event shows a CSC particle was preferentially delivered to a PM site that coincides with presence of a cortical microtubule and the particle later tracks along the cortical microtubule during the steady movement phase. Bar = 1 μm. (**B**) Representative single images show the PM of hypocotyl epidermal cells that were pre-treated with mock (0.1% DMSO) or 20 μM oryzalin for 2 h. Transgenic Arabidopsis lines expressing YFP-CESA6 or YFP-TUB5 were imaged with SDCM. Bar = 5 μm. (**C**) Representative single optical section images taken at cortical and subcortical focal planes in cells expressing YFP-CESA6 treated with mock or oryzalin for 2 h. The cytoplasmic CESA vesicles are highlighted with magenta circles. Bar = 5 μm. (**D**) Density of PM-localized CSCs measured in mock- or oryzalin-treated cells from images such as those shown in (**B**). Values given are means ± SE (n = 20–25 cells per treatment, Student’s *t*-test, nd: *p* > 0.05). (**E**) Density of small CESA vesicles in the cortical and subcortical cytoplasm measured from images such as those in (**C**). Values given are means ± SE (n = 20–22 cells per treatment, Student’s *t*-test, nd: *p* > 0.05).

**Figure 4 ijms-23-00317-f004:**
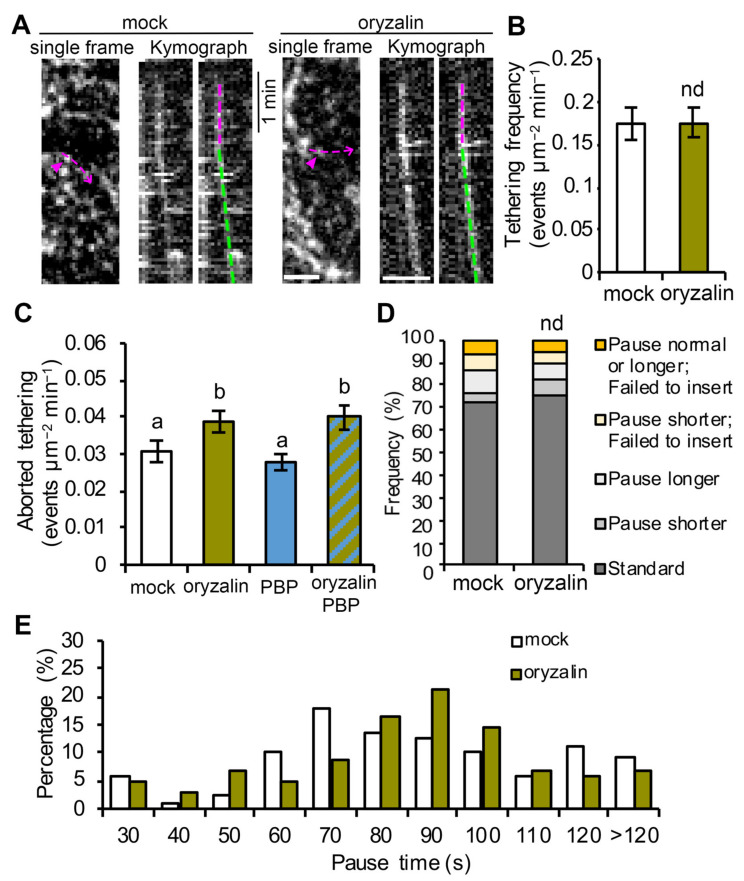
Cortical microtubules play a minor role in successful vesicle tethering or fusion of CSCs: (**A**) Representative kymographs of CSC insertion events in mock- or oryzalin-treated cells. The newly inserted CSC particle (magenta arrowheads) and its translocation trajectory (magenta dashed arrows) used to generate the kymograph are marked on the single frame image. The pause phase (magenta dashed lines) and the translocation phase (green dashed lines) of an insertion event are marked on the kymographs. Bars = 2 μm. (**B**) Analysis of CSC vesicle tethering frequency, as described in Figure 2, shows no significant difference between mock- and oryzalin-treated cells. Values given are means ± SE (n = 10–12 cells per treatment, Student’s *t*-test, *p* > 0.05). (**C**) Quantitative analysis shows increased frequency of aborted tethering events in oryzalin alone or combined oryzalin- and PBP-treated cells, compared with mock-treated cells. Values given are means ± SE (n = 15–20 cells per treatment, one-way ANOVA with Tukey’s post hoc test, *p* < 0.05). (**D**) Analysis of CSC insertion events and proportions of different types of insertion event in mock- or oryzalin-treated cells using the categories defined in Figure 1D. A total of 110 and 103 insertion events from 10 cells were measured in mock- and oryzalin-treated cells, respectively (chi-square test, nd: *p* > 0.05). (**E**) Distribution of pause times during CSC insertion at the PM for the insertion events shown in (**D**).

**Figure 5 ijms-23-00317-f005:**
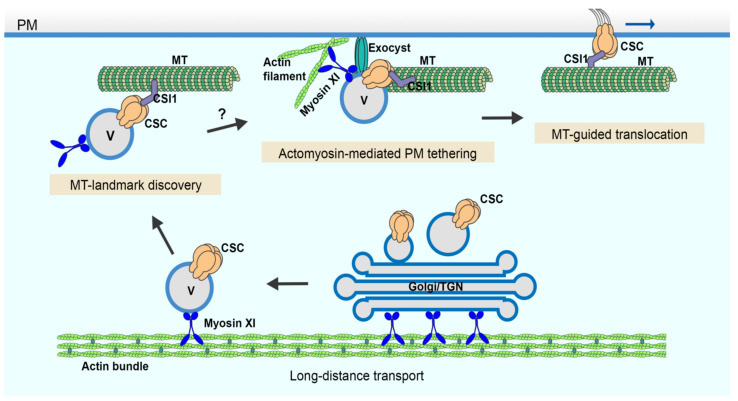
A cartoon showing the multiple and distinct roles of the cortical cytoskeleton in CSC secretion. CSC-containing Golgi/TGN and/or small CSC-containing vesicles are responsible for CSC delivery to the PM. These compartments undergo long-distance transport powered by myosin XI motors moving along actin filament cables or bundles located throughout the cytoplasm. In the cortical cytoplasm, CSC-containing vesicles associate with cortical microtubules potentially marking PM insertion sites. Tethering of the vesicle to the PM occurs either simultaneously, or slightly earlier or later than its association with cortical microtubules. Efficient vesicle tethering is achieved by the coordinated action of actin, myosin XI and the exocyst complex prior to subsequent membrane fusion and CSC insertion. Once inserted, CSCs translocate in the plane of the PM along trajectories defined by cortical microtubules while synthesizing cellulose microfibrils. MT: microtubule; V: vesicle.

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
