# Peer review of "Revising the Role of Cortical Cytoskeleton during Secretion: Actin and Myosin XI Function in Vesicle Tethering"

_ijms, 2021, doi:10.3390/ijms23010317_

Round 1
Reviewer 1 Report
In this manuscript the authors investigate the process of exocytosis by tracking delivery of the cellulose synthase complex to the plasma membrane. In particular they look at the role that myosin and actin play through the use of inhibitors or mutation. In addition, using the same platform, they assess the role that microtubules play in the same process.
Overall, this is a nice piece of work that builds on previous work by the authors. By using spinning disc confocal microscopy they are able to track single particles and how vesicle tethering is dependent on functional actin and myosin. In contrast, microtubules seem to play only a minor role in this process. I think that they have done a nice job showing this and I don’t have any serious concerns about the experimental design or the validity of the findings.
I do have a question about one aspect of the results. The authors have used a xik-2 mutant line and can phenocopy the effects of the mutation by using PBP. The effects of the PBP treatment are consistently more pronounced. Does this mean that multiple myosins are being targeted by the PBP? Is PBP also an inhibitor of myosin V? Regardless, I think the authors should address this somewhere in the text, since the mutation and inhibitor are not equivalent.
Minor points:
The legend for figure 1 has multiple fonts.
In figure 1 legend, include the number of cells per genotype or treatment.
Figure 4E – there is a shift toward a longer pause upon treatment with oryzalin. This is not consistent with the mean pause times that are shown in the figure.
The discussion is a bit repetitive and could be shortened.
Reviewer 2 Report
The Manuscript from Zhang and Staiger "Revising the Role of Cortical Cytoskeleton During Secretion: Actin and Myosin XI Function in Vesicle Tethering" is a nicely written and experimentally designed work.
Their findings are in agreement with the latest studies on the role of microtubule and actin on vesicle transport and secretion and I would suggest to add more references about this part.
